# XAC4296 Is a Multifunctional and Exclusive *Xanthomonadaceae* Gene Containing a Fusion of Lytic Transglycosylase and Epimerase Domains

**DOI:** 10.3390/microorganisms10051008

**Published:** 2022-05-11

**Authors:** Amanda C. P. de Oliveira, Rafael M. Ferreira, Maria Inês T. Ferro, Jesus A. Ferro, Caio Zamuner, Henrique Ferreira, Alessandro M. Varani

**Affiliations:** 1Graduate Program in Agricultural and Livestock Microbiology, School of Agricultural and Veterinary Sciences, São Paulo State University (UNESP), Jaboticabal 14884-900, SP, Brazil; amandacpoliveira@gmail.com; 2Department of Agricultural and Environmental Biotechnology, School of Agricultural and Veterinary Sciences, São Paulo State University (UNESP), Jaboticabal 14884-900, SP, Brazil; marini64@gmail.com (R.M.F.); maria.ferro@unesp.br (M.I.T.F.); jesus.ferro@unesp.br (J.A.F.); 3Graduate Program in Genetics and Plant Breeding, School of Agricultural and Veterinary Sciences, São Paulo State University (UNESP), Jaboticabal 14884-900, SP, Brazil; 4Department of General and Applied Biology, Institute of Biosciences, São Paulo State University (UNESP), Rio Claro 13506-900, SP, Brazil; c.zamuner@unesp.br (C.Z.); henrique.ferreira@unesp.br (H.F.)

**Keywords:** citrus canker, multi-domains enzymes, gene fusion, cell wall synthesis, antimicrobial resistance, *Xanthomonas citri*

## Abstract

Microorganisms have a limited and highly adaptable repertoire of genes capable of encoding proteins containing single or variable multidomains. The phytopathogenic bacteria *Xanthomonas citri* subsp. *citri* (*X. citri*) (*Xanthomonadaceae* family), the etiological agent of Citrus Canker (CC), presents a collection of multidomain and multifunctional enzymes (MFEs) that remains to be explored. Recent studies have shown that multidomain enzymes that act on the metabolism of the peptidoglycan and bacterial cell wall, belonging to the Lytic Transglycosylases (LTs) superfamily, play an essential role in *X. citri* biology. One of these LTs, named XAC4296, apart from the Transglycosylase SLT_2 and Peptidoglycan binding-like domains, contains an unexpected aldose 1-epimerase domain linked to the central metabolism; therefore, resembling a canonical MFE. In this work, we experimentally characterized XAC4296 revealing its role as an MFE and demonstrating its probable gene fusion origin and evolutionary history. The XAC4296 is expressed during plant-pathogen interaction, and the Δ4296 mutant impacts CC progression. Moreover, Δ4296 exhibited chromosome segregation and cell division errors, and sensitivity to ampicillin, suggesting not only LT activity but also that the XAC4296 may also contribute to resistance to β-lactams. However, both Δ4296 phenotypes can be restored when the mutant is supplemented with sucrose or glutamic acid as a carbon and nitrogen source, respectively; therefore, supporting the epimerase domain’s functional relationship with the central carbon and cell wall metabolism. Taken together, these results elucidate the role of XAC4296 as an MFE in *X. citri*, also bringing new insights into the evolution of multidomain proteins and antimicrobial resistance in the *Xanthomonadaceae* family.

## 1. Introduction

Many Gram-negative phytopathogens relevant to agriculture belong to the *Xanthomonadaceae* family [1]. A critical species in this group is the *Xanthomonas citri* subsp. *citri* (*X. citri*), the causal agent of citrus canker (CC) [2]. CC is a severe disease that affects citrus crops and decreases fruit production, leading to economic losses [3]. Many efforts to understand CC mechanisms have been made since the disease was discovered in the early 1900s [4]. One of the hallmarks that led to several new insights into the plant-pathogen interactions was the analyses of the *X. citri* genome, revealing the genetic basis of bacterial pathogenicity [5]. Since then, most studies have been focused on *Xanthomonas* pathogenicity mechanisms, such as regulation and secretion of virulence factors, such as the Type 3 Secretion System [5,6]. Moreover, genetics studies were also conducted to understand chromosome segregation and cell division mechanism, aiming to better understand this phytopathogen’s cellular biology [7,8]. However, other possible genetic mechanisms related to *X. citri* pathogenicity remain unknown. For instance, multidomain and multifunctional enzymes (MFEs) are essential for bacterial cellular biology, virulence, and fitness.

The MFEs are ubiquitous in prokaryotes [9]. These proteins generally harbor more than one domain, each exhibiting distinct functions [10]. Therefore, the MFEs may simultaneously perform multiple physiologically biochemical or biophysical tasks in the cell [11,12]. These numerous functionalities might provide evolutionary advantages for the bacterium [13]. For instance, combining multiple functions enables the enzyme to catalyze different steps of a single metabolic pathway [14]. In addition, the MFEs can be considered a clever strategy for generating complexity from existing proteins without expanding the genome [13].

Besides MFEs, one interesting class of enzymes has gained attention for their relation to bacterial fitness and virulence. These are the Lytic Transglycosylases (LTs) related to peptidoglycan biosynthesis and recycling and cell-wall-antibiotic detection, also showing involvement with the bacterium septum division allowing cell separation and insertion of protein complexes like secretion systems, flagella, and pili [15,16,17,18,19,20,21]. Due to these features, LTs may also play a relevant role in the pathogenesis of many bacterial species, such as *Neisseria gonorrhoeae* [22] and *Burkholderia pseudomallei* [23].

Recently, we described the LT’s arsenal present in the *X. citri* genome (16 LTs from different families) [24]. Among those, we functionally revealed that two LTs from the 3B family: MltB2.1 and MltB2.2, are directly implicated in *X. citri* fitness [24]. We also identified another 3B-like LT named XAC4296 (NCBI locus_tag: XAC_RS21660). Notably, apart from the Transglycosylase SLT 2 (IPR031304) and Peptidoglycan binding-like (IPR002477) domains, XAC4296 contains an additional and unexpected aldose 1-epimerase domain (IPR015443) linked to carbohydrate metabolism, and potentially showing involvement with the bacterial cell wall metabolism and biosynthesis of a variety of cell surface polysaccharides [25]. Interestingly, the XAC4296 gene was previously identified exclusively in the *Xanthomonas* genus [24]. Moreover, in silico analyses revealed that XAC4296 appears to have been formed by a previous gene fusion event, which originated by two independent genes (a 3B family LT and D-hexose-6-phosphate mutarotase gene), commonly separated in distinct loci in other non-*Xanthomonas* species [24]. Therefore, XAC4296 resembles a canonical MFE, showing a multidomain architecture.

In this work, we performed in silico, fluorescence microscopy and pathogenicity assays to investigate the evolution and role of XAC4296 as a putative MFE. We also evaluated XAC4296 as a potential *X. citri* virulence and pathogenicity factor. Our results indicate that XAC4296 functions resemble a typical LT, mainly related to peptidoglycan biosynthesis. We also unveiled an additional role related to carbohydrate metabolism, compatible with the epimerase domain’s role and chromosome segregation during cell division. Taken together, these results demonstrate that XAC4296 behaves like a classic MFE, showing at least two unrelated and mechanistically different roles, both impacting *X. citri* fitness, a primary role related to enzymatic catalysis and a secondary role related to cell structural function.

## 2. Material and Methods

### 2.1. In Silico and Phylogenetic Analysis

Global comparisons were made based on sequenced *Xanthomonadaceae* genomes deposited in the National Center for Biotechnology Information (NCBI) repository. For XAC4296 homolog detection, we used as tblastn [26] parameters a query coverage and identity >90% and >60%, respectively, and including all three characteristics domains (Appendix A shows a complete list of genomes carrying XAC4296 homolog).

The XAC4296 three-dimensional structure was analyzed in two modules, the first considering the 420 amino acids located in the N-terminus of the protein, containing the SLT_2 (IPR031304) and PG_Binding 1 (IPR002477) domains, corresponding to the Lytic Transglycosylases from 3B family; the second module considering the last 309 amino acids located at the C-Terminus of the XAC4296 and corresponding to the D-hexose-6-phosphate mutarotase annotated gene, containing the aldose-1-epimerase domain (IPR015443). Molecular modeling was performed with the Robetta webserver [27]. The Chimera Tool [28] was used to generate the three-dimensional structures and interactive visualization of the entire XAC4296 protein. The three-dimensional structures of the PDB (Protein Data Bank) used as models for the LT and epimerase modules were, respectively, 5AO8 and 2HTA. These proteins were selected according to the ranking established by the alignment performed by software MAFFT 7309 [29]. The stereochemical quality of the generated models was evaluated by analyzing Ramachandran’s plot, carried out by Chimera Tool.

The XAC4296 homolog sequences were aligned with MAFFT 7.309 [29], and their best-fit evolutionary models were predicted with ProTest 3.2.4. A maximum-likelihood tree was reconstructed with RaxML 8.2.9 using a bootstrap value of 1000. The final tree was visualized in FigTree 1.4.4 (http://tree.bio.ed.ac.uk/software/figtree (accessed on 20 November 2021)) and edited with Inkscape 0.92.4 (http://www.inkscape.org (accessed on 20 November 2021)). The Integrated Microbial Genomes & Microbiomes (IMG/M) system [30] was used for comparative analyses.

### 2.2. Strains and Growth Conditions

Bacterial strains and plasmid strains used in this study are shown in Appendix A. The *X. citri* strains were grown in three different culture media: nutrient broth (NB: 0.5% peptone, 0.3% beef extract), nutrient agar (NA: 0.5% peptone, 0.3% beef extract, 0.15% agar) supplemented with L-arabinose (0.05% *w*/*v*) and sucrose (5% *w*/*v*) when required or, XVM2 (20 mM NaCl, 10 mM (NH_4_)_2_SO_4_, 5 mM MgSO_4_, 1 mM CaCl_2_, 0.16 mM KH_2_PO_4_, 0.32 mM K_2_HPO_4_, 0.01 mM FeSO_4_, 10 mM fructose, 10 mM sucrose, 0.03% casaminoacids, pH 6.7) at 29 °C. *Escherichia coli* strains were cultivated in Luria-Bertani medium (LB: 1% tryptone, 0.5% yeast extract, 0.10% NaCl, 0.15% agar; pH 7) and SOB media [31] at 37 °C. Antibiotics were used as needed at the following concentrations: kanamycin (Kn), 30 µg/mL; carbenicillin (Carb), 50 µg/mL; streptomycin (Str), 50 µg/mL; gentamycin (Gen), 10 µg/mL; ampicillin (Amp), 100 µg/mL.

### 2.3. RNA Extraction and cDNA Synthesis from XAC4296

*X. citri* total RNA was extracted using RNeasy protect bacteria Mini kit (Qiagen) according to the manufacturer. The first strand of complementary DNA was synthesized from 1 μg of total RNA using a qScript^®^ cDNA SuperMix (Qiagen). Before cDNA synthesis, RNA samples were treated with DNaseI. The DNA and RNA quantification was performed using Qubit HS (High Sensitivity) (Thermo Fisher, Waltham, MA, USA). Primers F4296 (F) and pMAJIIc (R) were used for PCR reaction using cDNA as a template (Appendix A). PCR products were checked by agarose gel electrophoresis.

### 2.4. Mutant Construction

Mutant of gene XAC4296 was generated using homologous suicide plasmid (pNPTS138) [32] integration through site-directed mutagenesis by PCR overlap extension approach [33]. To construct the deletion mutant of XAC4296 ORF, we used *X. citri* genomic DNA as a template and primers described in Appendix A. The first PCR amplifications were made separately using pairs of primers A(F)–B(R) and C(F)–D(R) and Phusion high fidelity DNA polymerase (Thermo Fischer Scientific) to generate the products A–B and C–D, with self-complementary tails. The second PCR was performed using primers A(F)–D(R) and the products A–B and C–D as a template to obtain the A–D fragment, in which the XAC4296 sequence was deleted. The final PCR product A–D and pNPTS138 suicide vector were double digested with *Nhe*I/*Hind*III enzymes (New England BioLabs Inc.^®^, Ipswich, MA, USA). The ligation between vector and fragments was performed with T4 DNA Ligase (New England BioLabs Inc.^®^) according to the manufacturer’s instructions. The recombinant vector Δ4296-pNPTS138 was transformed into chemically competent *E. coli* DH10B [31], and transformant colonies were selected using antibiotics and Lac-Z promoter. The constructions were checked by agarose gel electrophoresis and sequencing on a 3730xI DNA analyzer (Thermo Fisher Scientific) using primers A–D. Finally, the Δ4296-pNPTS138 recombinant plasmid was used for *X. citri* electroporation [34], and colonies were selected by kanamycin resistance and sucrose susceptibility [35]. Mutant Δ4296 was confirmed by sequencing.

The XAC4296 ORF was PCR amplified from *X. citri* genomic DNA using primers pMAJIIc (F)-pMAJIIc (R) (Appendix A) and Phusion high fidelity DNA polymerase (Thermo Fisher Scientific). The PCR product and the integrative vector pMAJIIc [36] were double digested with *Nhe*I/*Xho*I enzymes (New England BioLabs Inc.^®^) and ligated with T4 DNA Ligase (New England BioLabs Inc.^®^), according to manufacturer’s instructions. The recombinant vector 4296-pMAJIIc was transformed into chemically competent *E. coli* DH10B [31]. Colonies were selected using kanamycin resistance. The recombinant plasmid DNA (pMAJIIc-XAC4296) was purified using Promega Wizard^®^Plus SV Minipreps DNA Purification System kit according to the manufacturer’s instructions and the inserted XAC4296 DNA sequence was confirmed by sequencing. The recombinant plasmid was used to transform both mutant strain 4296 and *X. citri* wild type strain by electroporation [34]. Colonies were selected by kanamycin resistance, and the integrative vector version was identified on NA plates supplemented with 0.2% soluble starch followed by iodine vapor crystals exposure [36]. The strain Δ4296-pMAJIIc-4296 (named Δ4296c) was used as complemented strain in the following assays, and the recombinant strain XccA-pMAJIIc-4296 was used for protein subcellular localization. The constructions were checked by agarose gel electrophoresis and DNA sequencing.

### 2.5. Pathogenicity Assay

We used two methods for pathogenicity assays. In the first method, bacterial strains were inoculated into the leaves’ surface only by spray [37]. *X. citri* and mutant Δ4296 were cultivated in NB medium for 16 h to O.D. 600-nm ~0.8 and diluted in fresh NB medium to O.D. 600 nm of 0.3. Cells were collected by centrifugation and resuspended in autoclaved tap water to an O.D. 600 nm of 0.3, equivalent to 10^8^ CFU/mL. Three different “Pêra Rio” orange (*Citrus sinensis* L. Osbeck) plants were sprayed with each bacterial suspension until all leaves were thoroughly coated, then covered with a clear plastic bag for 24 h. After 25 days of inoculation (DAI), all leaves were quantified, those presenting citrus canker (CC) symptoms were photographed, the CC lesions were counted, and the results were analyzed and compared.

For pathogenicity assays by infiltration method, strains of *X. citri* and mutant Δ4296 were cultivated in NB medium for 16 h to O.D. 600 nm ~0.8 and diluted in fresh NB medium to O.D. 600 nm of 0.3. Cells were collected by centrifugation and resuspended in autoclaved tap water to an O.D. 600 nm of 0.3, equivalent to 10^8^ CFU/mL. This inoculum was diluted 100-fold (10^6^ CFU/mL) and infiltrated on the abaxial surface of three young leaves (technical replicates) in three different plants (biological replicates) of “Pera Rio” orange (*C. sinensis* L. Osbeck) using 1 mL needleless hypodermic syringes [38]. Symptoms were observed for 25 days, and photos were taken at 4, 8, 12, 15, and 21 days after inoculation (DAI) [38].

Inoculated plants were kept in a high-efficiency particulate air (HEPA) filtered plant laboratory with controlled environmental conditions (28–30 °C, 55% humidity, 12 h light cycle).

### 2.6. Ex Vivo Growth Curves

*X. citri* and Δ4296 mutant were cultivated in NB medium for 16 h and diluted in fresh NB medium to O.D. 600-nm of ~0.1. Cell cultures were distributed on 96 well plates and were incubated in a Synergy H1N1 microplate reader (BioTek^®^, Winooski, VT, USA) under constant agitation at 29 °C, and automated O.D. readings were taken every 30 min. Using GraphPad Prism 6 software, growth curves were generated based on three technical and three biological replicates [7].

### 2.7. In Planta Growth Curves

*X. citri* and Δ4296 mutant were cultivated in NB medium for 16 h until O.D. 600 nm got around 0.8 and diluted in fresh NB medium to O.D. 600 nm of 0.3. Cells were collected by centrifugation and resuspended in Falcon tubes containing 50 mL of autoclaved tap water to an O.D. 600 nm of 0.3, equivalent to 10^8^ CFU/mL. This inoculum was diluted 100-fold (10^6^ CFU/mL) and infiltrated on the abaxial surface of fifteen young leaves in four different plants (biological replicates) of “Pera Rio” orange (*C. sinensis* L. Osbeck) using 1 ML needleless hypodermic syringes. The strains were exuded from leaves at days 0, 1, 3, 6, and 10 DAI, and the number of cells per leave was achieved using the microculture strategy [38].

### 2.8. Microscopy

*X. citri*, Δ4296 mutant, and Δ4296c strains were cultivated in NB media until O.D. 600-nm reached around 0.3 ABS at 29 °C. We performed analysis in different conditions: with ampicillin (20 µg/mL), sucrose 2% (*w*/*v*) and glutamic acid 2% (*w*/*v*). For morphological analysis, strains were collected by centrifugation, and cells were resuspended in 0.85% NaCl. We used 4′,6-diamidino-2-phenylindole DAPI staining at a final concentration of 0.01% to visualize chromosome organization. Cultures were treated with propidium iodide (IP) for cell viability investigations at a 0.001 mg/mL final concentration. Cells were immobilized in agarose-covered slides for microscope observation [39]. We performed the assays three times and quantified cells individually (*n* = 800). Following treatments, cells were immediately visualized using an Olympus BX61 microscope equipped with a monochromatic camera OrcaFlash 2.8 (Hamamatsu, Japan). The software CellSens Version 11 (Olympus, Tokyo, Japan) was used for data collection and analysis.

### 2.9. Data Analysis

All results obtained were submitted to Welch’s ANOVA test-0.05 using GraphPad Prism 8.0.1. Graphics were generated using Microsoft Office Excel for Windows.

## 3. Results

### 3.1. XAC4296 Is Conserved in Xanthomonas, Pseudoxanthomonas, and Stenotrophomonas, Showing Two Distinct and Independent Domains Modules

We first analyzed the XAC4296 gene (2163 bp) using molecular modeling approaches. The Robetta software failed to generate a unique 3D structure corresponding to the complete protein containing LT and epimerase domains. Therefore, both domains were modeled separately to unravel the XAC4296 (720 aa) structure. While the LT located on the N-terminus of XAC4296, containing the SLT_2 and PG_Binding 1 domains, was modeled based on a sequence of 408 amino acids, the XAC4296 C-terminus, having the epimerase bearing the aldose-1-epimerase domain, was modeled based on 312 amino acids sequence (Figure 1A,B).

The first module resembles a classic 3B LT showing the 3D structure composed mainly of alpha-helices (Figure 1C). In contrast, the second module shows mainly beta-sheets, exhibiting high identity to other well-characterized D-hexose-6-phosphate mutarotase (Figure 1C). Therefore, it is presumed that the aldose 1-epimerase domain might perform the epimerization by ring-opening or mutarotation [40]. The confidence value for both models was 0.86 and 0.81, respectively, supporting a reasonable quality of both 3D structure predictions. These findings suggest that XAC4296 may have two completely independent domains that might preserve the transglycosylase and epimerase activities separately.

We also investigated the origin and evolution of XAC4296, revealing that apart from different species of *Xanthomonas*, this gene can also be found in different members of the *Xanthomonadaceae* family, such as *Pseudoxanthomonas* and *Stenotrophomonas*, accounting for at least 308 complete sequenced genomes available in the GenBank database (Figure 2A and Appendix A). Moreover, considering these three genera, the XAC4296 homologs are generally located in a conserved genomic context (Appendix A). The XAC4296 and their homologs are not associated with common mobile genetic elements, such as prophages, Insertion Sequences, Transposons, Integrons, and Genomic Islands. Therefore, strongly supports that the XAC4296 origin is not associated with common lateral gene transfer mechanisms. It is noteworthy to mention that the XAC4296 homolog is not present in the *Xanthomonas albilineans* and the phylogenetically closely-related *Xylella* genus (Figure 2A). *Xylella* carries an independent epimerase gene (i.e., WP_010894718.1) and a least 4 LTs genes, all located at distant genomic loci. Conversely, in addition to an aldose 1-epimerase domain-containing gene (i.e., XALC_0947 and XaFJ1_GM000925), the *X. albilineans* species (e.g., GPE PC73 and Xa-FJ1 strains), also carries the two modules (LT and Epimerase) as separated and overlapping genes (21 nucleotides of overlapping), each gene showing the domains modules in different frames, and resembling a degenerated XAC4296 homolog (Appendix A). Moreover, the other *Xanthomonas* species, *Pseudoxanthomonas* and *Stenotrophomonas* carry their own set of LTs (varying in number and diversity of families). However, the aldose-1-epimerase domain is exclusive for each XAC4296 homolog and thus does not exist as an alone or duplicated feature such as the D-hexose-6-phosphate mutarotase gene in these genomes, as observed in *Xylella* and *X. albilineans*. 

### 3.2. X. citri Expresses XAC4296 in Citrus sinensis L. Osbeck, and Δ4296 Mutant Does Not Affect Bacterial Growth but Impacts the CC Progression

We further investigated the role of XAC4296 on *Xanthomonas citri* virulence and pathogenicity. To confirm the XAC4296 expression during plant-pathogen interaction, *Citrus sinensis* L. Osbeck (considered a moderately resistant host) was inoculated with *X. citri*. The pathogen was exuded five days after inoculation. The synthesis of double-stranded cDNA from total RNA was performed and used as a template for PCR reaction. The fragment with approximately 2200 pb was obtained, indicating that XAC4296 is entirely expressed during plant-pathogen interaction (Appendix A). 

Furthermore, the XAC4296 gene was deleted by site-directed mutagenesis to generate mutant *X. citri* Δ4296, and the integrative plasmid pMAJIIc [36] was employed to construct the complemented strain Δ4296c. Different bacterial inoculation methods (spray and infiltration) (Table 1 and Appendix A) and growth curve analyses (Appendix A) indicated that the wild-type and mutant exhibited similar growth patterns; pointing to the fact, however, that XAC4296 can be related to CC symptoms enhancement in different hosts (e.g., *Citrus latifolia* Tan, *Citrus sinensis,* and *Citrus reticulata*) regarding only the spray method. 

### 3.3. Δ4296 Cells Displayed Abnormal Nucleoid Distribution, Chains, and Short Filaments

To evaluate the potential role of the XAC4296 LT domain with bacterial *Peptidoglycan* metabolism, the cell morphology and chromosome organization of wild-type *X. citri* and the Δ4296 mutant were investigated using DAPI-staining (Figure 3 and Table 2). The wild-type *X. citri* displays an average cell length of approximately 1.44 ± 0.31 µm [8]. Moreover, in standard growth conditions, *X. citri* exhibits a bilobed nucleoid centrally located in a single compartment (newborn or not-dividing rods) (Figure 3A) or evenly distributed, one per cell half, when division constriction is present. 

Different from the wild type, the Δ4296 mutant formed chains and sometimes a phenotype resembling short filaments, accompanied by irregular distribution of the chromosomal mass (Figure 3B, white arrows). The proportions of short filaments and chains measured in a given mutant culture were 30.6% and 15%, respectively (Table 3). Close inspection of the chains with clear septal constrictions (red arrows in Figure 3) showed no apparent nucleoid bisection. However, the Δ4296 mutant seems perfectly competent in chromosome segregation because many cells in a culture display normal nucleoid distribution (Figure 3B). However, in successive cellular cycles, errors in late cell division lead to the accumulation of chromosomal mass generated by subsequent events of replication. The short filaments morphotype that happened nearly twice as much as the chains could be caused by triggering the division error in an early stage of the cell cycle or converting chains into short filaments. At the moment, we cannot pinpoint which of these options is taking place. Finally, complementation of *X. citri* Δ4296 with pMAJIIc (Δ4296c) expressed from an ectopic site completely restored the wild-type phenotype (Figure 3C).

These findings suggest that the first cell error probably occurs mainly during late cell division. The filamentation seems to happen due to the cell losing the ability to form septa in successive cellular cycles.

### 3.4. XAC4296 Is Required for Proper β-Lactam Resistance and May Affect Cell Wall Synthesis

Considering that other members from the *Xanthomonas* genus are naturally resistant to ampicillin [42] (in *X. citri* the XAC_RS19350/XAC3833 gene provides the resistance), and the cell wall synthesis is closely associated with cell morphology, we further evaluated whether the XAC4296 deletion would affect the sensitivity to ampicillin, the cell morphology and the chromosome organization of wild type *X. citri* and the Δ4296 mutant in the presence of ampicillin antibiotic using DAPI-staining (Figure 4 and Table 3). The Δ4296 mutant showed a significant growth delay in the presence of ampicillin, suggesting that antibiotic sensitivity is pronounced (Figure 4A). All cells of the Δ4296 cultures displayed short filaments and chains with the abnormal distribution of the nucleoids (Figure 4B and Table 3), indicating an altered or disturbed nucleoid segregation. Permeability analysis showed that membrane disruption was not detected (Appendix A). 

### 3.5. XAC4296 Is Related to Bacterial Central Carbon Metabolism

To evaluate the potential role of the XAC4296 epimerase domain, two different carbon and nitrogen sources were used to supplement the NB medium with ampicillin. In the presence of sucrose, the Δ4296 mutant cell shape and chromosome organization were partially restored (Figure 5 and Table 4), but cells still showed short filaments, chains, and unusual chromosome organization/segregation (Figure 5A,B and Appendix A, Table 4). In contrast, the addition of glutamic acid-enhanced Δ4296 growth to the point where it matches that of the wild type *X. citri* and fully reversing back to normal chromosome organization/segregation and cell division (Figure 6A,B and Appendix A, Table 4). Considering that: (a) D-hexose-6-phosphate mutarotase gene containing the aldose 1-epimerase domain can be related to epimerization by ring-opening or mutarotation acting in the central carbon metabolism and impacting the cell-wall metabolism, and (b) the supplementation with different carbon sources restores the Δ4296 mutant to the wild-type *X. citri* phenotype; these results support that the XAC4296 epimerase domain might have an essential relation with the central carbon metabolism.

## 4. Discussion

In this work, we investigate the role of the XAC4296 in *Xanthomonas citri* metabolism and virulence using *in silico* and molecular approaches. The XAC4296 protein contains two distinct modules: the first containing the SLT_2 (IPR031304) and PG_Binding1 (IPR002477) domains, homologs to LTs 3B Family. The second module contains the aldose-1-epimerase domain (IPR015443), classified as an epimerase superfamily. Proteins belonging to the LTs (Lytic Murein Transglycosylases) family cleave the polysaccharide of the peptidoglycan at the NAM-NAG glycosidic bond by intramolecular cyclization of the n-acetylmuramyl moiety to yield a 1,6-anhydro-n-acetyl-β-d-muramyl (1,6-anhydroMurNAc) product during the peptidoglycan biosynthesis [16]. On the other hand, epimerases are usually involved in metabolic pathways such as inversion of D-alanine and D-glutamate for bacterial cell wall metabolism [25]; biosynthesis of a variety of cell surface polysaccharides; biosynthesis of LPS and capsular sugar precursors [43]; and complex biosynthetic pathways, such as Glycolysis, Entner-Doudoroff, Leloir and others that present several chemical steps [44,45]. Epimerases are also involved in oxidation, acetylation, dehydration, and carbohydrate reduction (reviewed by [40]). Interestingly, we did not find any difference in plate colony appearance or biofilm production of the Δ4296 mutant compared to the *X. citri* wild type strain, indicating that, in principle, the XAC4296 epimerase domain is not related to cell surface polysaccharides production (data not shown). Indeed, the presented results indicated that the XAC4296 epimerase domain might be associated with other metabolic pathways, such as those related to bacterial cell wall metabolism. Our molecular modeling results also support that the LT and epimerase modules can act independently and may synergistically function as a canonical MFE, thus, performing multiple physiologically biochemical or biophysical functions simultaneously in the cell. 

Previous studies revealed that the XAC4296 modules exist as separate and independent genes in other γ-proteobacteria [24]. Our results indicate that XAC4296 homolog is widespread in different *Xanthomonas* species and exists in other members of the *Xanthomonadaceae* family, such as *Stenotrophomonas* and *Pseudoxanthomonas*. It is worth mentioning that the *Xanthomonas, Stenotrophomonas, Xylella,* and *Pseudoxanthomonas* genera from the *Xanthomonadaceae* family are closely related and form a phylogroup [41]. Therefore, our results support that the XAC4296 origin may be related to a previous gene fusion origin before or during this *phylogroup differentiation*. However, *Xylella* and *Xanthomonas albilineans* do not have the XAC4296 homolog. For instance, *Xylella* carries an independent D-hexose-6-phosphate mutarotase gene containing the aldose-1-epimerase domain, and their LT gene repertoire is in a distinct genomic locus. In contrast, the *X. albilineans* show both protein modules, the LT and epimerase, homologs to XAC4296, as separate but overlapping genes and an additional and independent D-hexose-6-phosphate mutarotase gene. It is known that the *Xylella* genus has undergone drastic genome reduction since diverging from the *Xanthomonas* genus [46,47] and that *X. albilineans* has experienced significant genomic erosion, having unique genomic features in comparison to other *Xanthomonas* species [48]. Therefore, the most parsimonious hypothesis to explain the absences of XAC4296 homologs in *Xylella* and *X. albilineans* is that while *Xylella* lost their XAC4296 homolog during the genome reduction process, but maintaining their epimerase gene alone, and their own LTs repertoire, the *X. albilineans* genome is accumulating mutations and nucleotide deletions that led to the formation of two separated genes, and thus, suggesting a current process of gene decay. This hypothesis is based on the current knowledge that the genome reduction or erosion process is currently shaping *Xyella* and *X. albilineans* and adapting both pathogens to a restricted host range [46,47,48]. 

This work further evaluated XAC4296 role in *X. citri* fitness, virulence, and pathogenicity. Our results indicate that the XAC4296 gene is not essential for *X. citri* survival or *in planta* CC development but may play a role in bacterial fitness. Our findings also support XAC4296 direct relationship with *X. citri* pathogenicity, particularly CC progression. However, this result is not surprising since previous studies revealed a role of the other LT from the 3B family with *X. citri* pathogenicity and fitness [24]. 

To explore the involvement of the XAC4296 LT domain with the peptidoglycan metabolism, the cell morphology was examined under the microscope using DAPI-staining. *X. citri* lacking XAC4296 formed chains related to late cell division errors. In these cells, the early division is normal, forming constriction but does not progress until complete closure of the septum and cell separation. This is not an unexpected result since many studies based on LTs mutants (from different families) reported similar phenotypes [22,49,50,51]. Indeed, cell-wall biosynthesis stays in a homeostatic balance between construction and demolition [52]. In the absence of the activity of the LT domain provided by XAC4296, the peptidoglycan maintenance is perturbed, leading to these morphological defects observed in *X. citri*. This finding supports the hypothesis that the XAC4296 function is related to the 1,6-anhydroMurNAc-containing muropeptides production, the hallmark of LT catalysis [16]. These muropeptides may be transported from the periplasm to the cytoplasm through the transmembrane protein AmpG [53]. Next, these muropeptides are degraded in the cytoplasm, and their components are used for Lipid II biosynthesis that is assembled in the cytoplasm and, again, delivered to the periplasm for de novo *s*ynthesis of the peptidoglycan [54,55,56,57,58,59,60,61,62,63,64]. Therefore, our results support the XAC4296 LT role, acting in the peptidoglycan synthesis and dynamics, consequently influencing cell shape as previously described for this class of enzymes [16]. 

In this work, we observed that the short filaments phenotype intensified when ampicillin was added to the Δ4296 mutant culture, impacting cell growth. Ampicillin is a β-lactam antibiotic that blocks the activity of penicillin-binding proteins [65]. Although other *Xanthomonas* members (e.g., *Xanthomonas campestris*) appears to express β-lactamase constitutively [42], in the Δ4296 strain, the absence of XAC4296 protein seems to hinder their ability to reconstruct cell wall making the bacteria susceptible to this antibiotic. Indeed, the bacterial exposure to β-lactams leads to disturbs in peptidoglycan recycling and accumulation of MurNAc-peptides in the cytoplasm [66]. In addition, during LT catalysis, the product 1,6-anhydroMurNAc-containing muropeptides are transported from the periplasm to the cytoplasm [54]. These muropeptides are transported through the transmembrane protein AmpG [54]. The muropeptides may be metabolized further through multiple routes to yield UDP-MurNAc-pentapeptide, a precursor of peptidoglycan biosynthesis [21,67], or may bind to AmpR and convert it into an activator of *ampC* transcription [66,68]. Here, we hypothesize that the observed behavior of Δ4296 may be explained due to the bacterial decreasing of the pool of muropeptides in the periplasm, consequently in the cytoplasm, leading to peptidoglycan synthesis imbalance and interrupting *ampC* transcription, and consequently, increasing β-lactams susceptibility. Therefore, our results strongly support that XAC4296 protein may also contribute to β-lactam antibiotic resistance in *X. citri*.

However, the morphological defects observed in the XAC4296 mutant (with or without the addition of ampicillin) can be directly related to the epimerase domain since our results also indicate that the late cell division errors observed in the chain and short filaments phenotypes can be wholly restored with sucrose and glutamate supplementation. One possible way to interpret these results is that the XAC4296 epimerase domain may contribute to many reactions in carbohydrate metabolism, such as the D-hexose-6-phosphate mutarotase. Its absence leads to an imbalance of metabolic precursors related to anabolism pathways, indirectly affecting bacterial cell division and chromosome segregation in *X. citri*. In the absence of XAC4296, glutamate and sucrose supplement may probably provide substrate for alpha-glucose-6-phosphate production for the central carbon source metabolism, and together with the activity of the other LTs from the 3B family present in the *X. citri* genome and their known functional redundancy mechanism [16,24] restored the bacterial metabolism and, consequently, the cell cycle. In addition, the glucose-6 phosphate is the primary metabolic substrate present for the glycolysis, phosphogluconate, and Entner–Doudoroff pathways. Nonetheless, the glutamate and sucrose supplement may provide a comprehensive carbon source to the bacteria, facilitating energy metabolism and bacterial adaptation and survival to stressful conditions, free-living, and wide host ranges. However, further studies such as protein purification and *in vitro* tests to check XAC4296 enzymatic activity are still needed to determine its role thoroughly.

Finally, previous studies raised the possibility of combining an LT inhibitor with β-lactam antibiotics as an alternative for future antibiotic development [69]. For example, the possible interaction between the LT Slt35 (3B family) and Bulgecin A [70], a potent LTs inhibitor that restores the efficiency of β-lactam antibiotics against resistant bacteria [69]. Taken together, our results strongly support these previous studies, suggesting the future development of 3B LT and epimerase inhibitors as potential new tools, not only for the control of the disease caused by *Xanthomonas* and other phytopathogens but also for antimicrobial resistance in general.

## 5. Conclusions

The XAC4296 protein seems to be related to late cell division leading to chromosome segregation errors and ampicillin susceptibility. Moreover, we have shown that XAC4296 mutants display a metabolism-dependent phenotype, resulting from the imbalance of metabolic precursors related to anabolism pathways, suggesting that XAC4296 also acts in central carbon metabolism. In conclusion, our results strongly indicate that XAC4296 encodes a multifunctional protein, playing a role as a transglycosylase as much as an epimerase, impacting bacterial fitness and bringing new insights into *X. citri* and other *Xanthomonadaceae* metabolism, evolution, and antimicrobial resistance emergence.

## Figures and Tables

**Figure 1 microorganisms-10-01008-f001:**
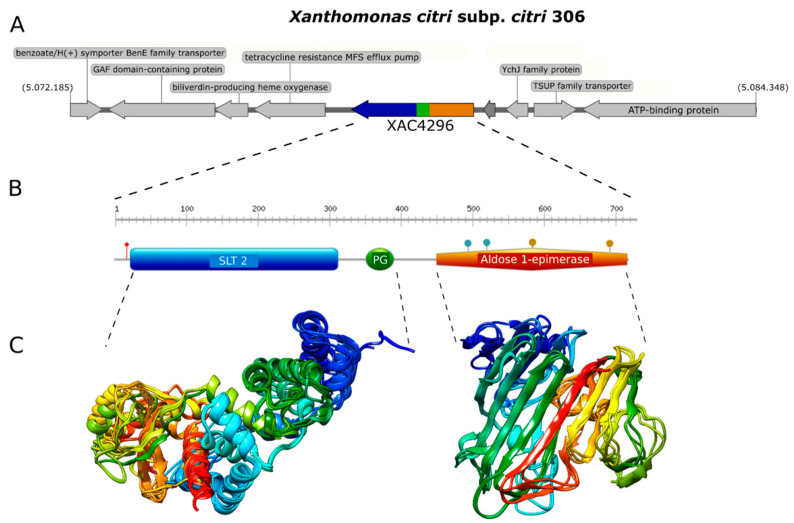
Genome context, protein domain, and structure of XAC4296. (**A**). Genome context of XAC4296 from *X. citri* genome. (**B**). Protein domain and structure of XAC4296. XAC4296 has 720 aa with the LT domain associated with 3B family: Transglycosylase SLT domain (SLT_2) (IPR031304.) and Peptidoglycan binding (PG_binding_1) (IPR002477) domains, and the Aldose 1-epimerase (IPR015443) domain. (**C**). Molecular modeling cartoon representation of the LT and epimerase XAC4296 domains.

**Figure 2 microorganisms-10-01008-f002:**
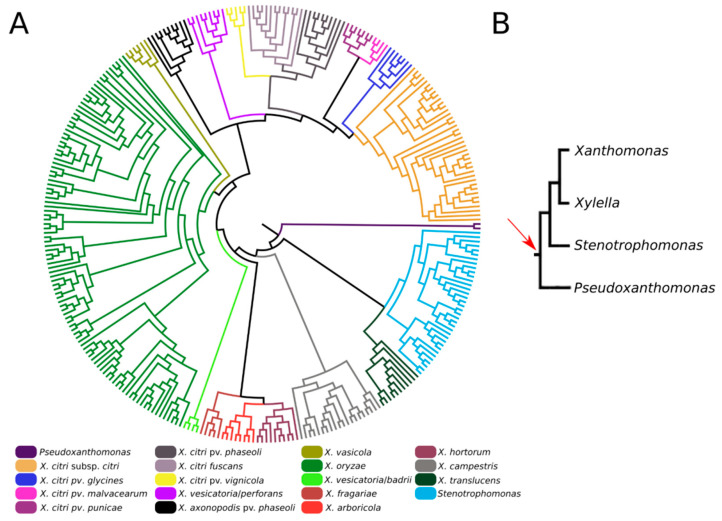
(**A**). Maximum-likelihood phylogenetic tree of XAC4296 homologs across the *Xanthomonadaceae* family supports the XAC4296 potential origin before *Xanthomonas*, *Xylella*, *Pseudoxanthomonas*, and *Stenotrophomonas* differentiation. The closely-related *Xylella* genus lost the XAC4296 homolog. (**B**). Phylogenetic construction of the *Xanthomonadaceae* family phylogroup formed by *Xanthomonas*, *Xylella*, *Pseudoxanthomonas*, and *Stenotrophomonas* (based on [41]). The red arrow indicates the potential gene fusion event that originated the XAC4296 ancestor.

**Figure 3 microorganisms-10-01008-f003:**
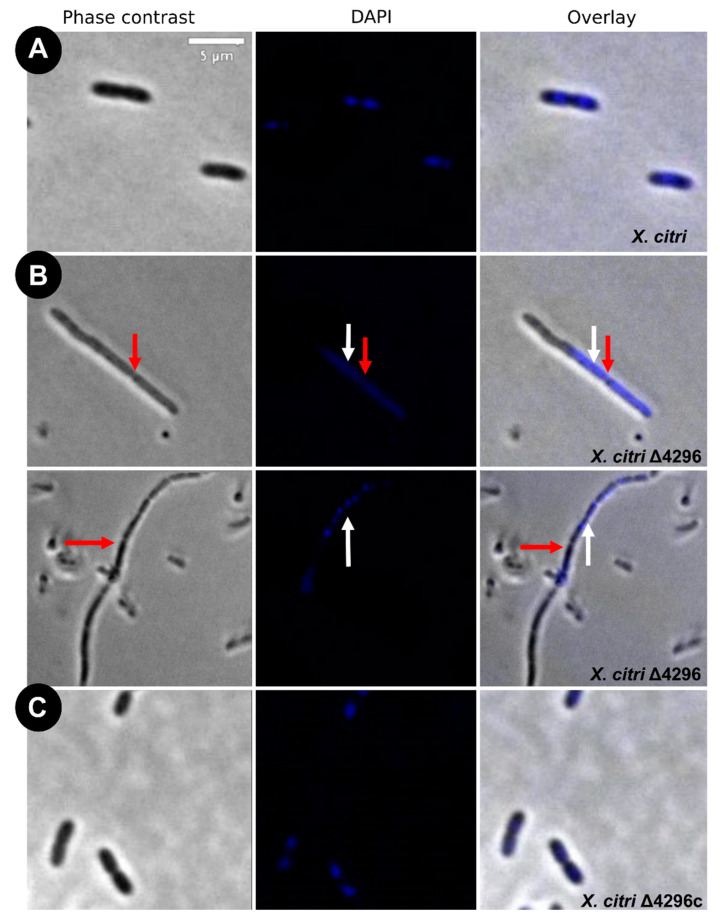
Morphological analysis and errors/aberrant nucleoid distribution of *X. citri*, Δ4296, and Δ4296c strains. Short filaments, chain phenotype, and errors/unusual nucleoid distribution were intensified without XAC4296. The figure shows microscopy phase contrast, DAPI, and overlay of the two filters for (**A**) *X. citri* WT; (**B**) Δ4296, and (**C**) Δ4296c (pMAJIIc-XAC4296) (magnification of 100×). White arrows indicate nucleoid distribution; red arrows indicate septum constriction (magnification of 100×)—scale = 5 µm.

**Figure 4 microorganisms-10-01008-f004:**
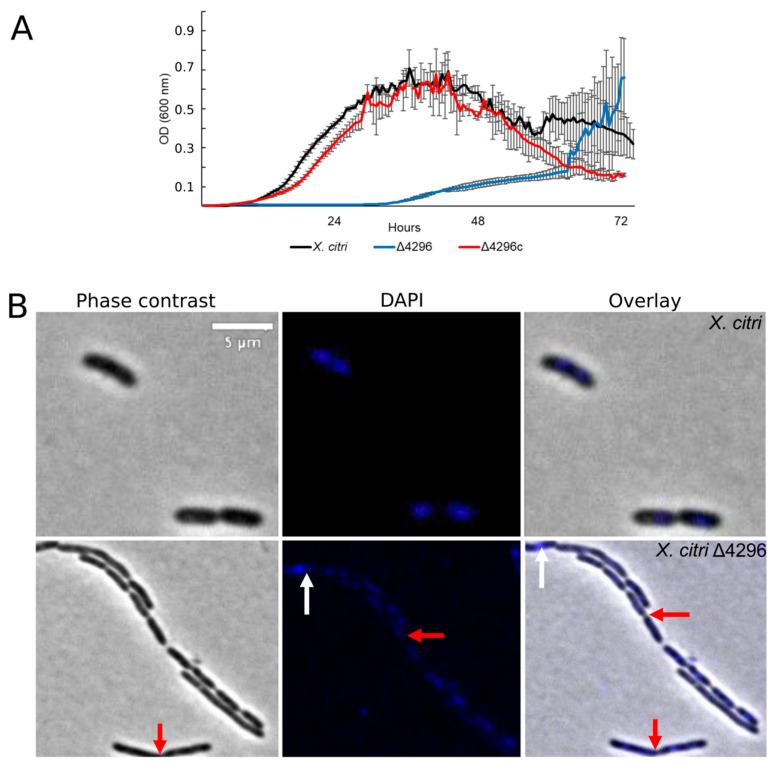
XAC4296 may be related to β-lactam antibiotic resistance in *X. citri* (**A**) Ex planta bacterial growth curves performed on rich medium and ampicillin 20 µg/mL for *X. citri*, Δ4296, and Δ4296c. Δ4296 growth is affected in the presence of ampicillin. Error bars indicate the standard error of three independent biological and technical replicates. (**B**) Morphological analysis and errors/aberrant nucleoid distribution of *X. citri*, Δ4296, and Δ4296c strains of *X. citri* and Δ4296 strains on NB supplemented with ampicillin 20 µg/mL. All cultures of mutant Δ4296 exhibit short filaments and chain phenotype. The figure shows the phase contrast, DAPI, and overlay of the two images for *X. citri* and Δ4296 (magnification of 100×). White arrows indicate the chromosome position, and red arrows indicate septum constriction (magnification of 100×)—scale = 5 µm.

**Figure 5 microorganisms-10-01008-f005:**
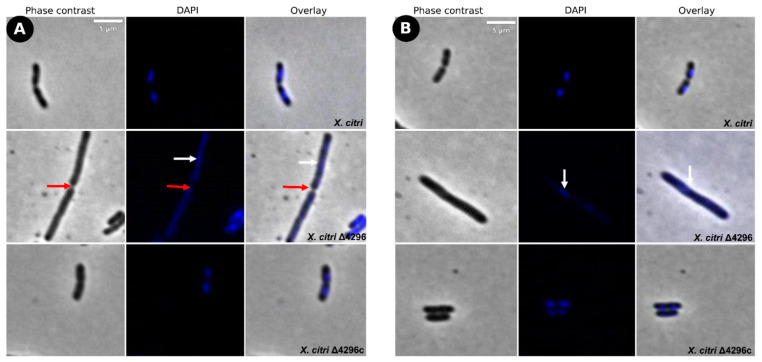
Short filaments and chain phenotype show partial reversion in sucrose’s presence as carbon source (**A**) Morphological analysis of *X. citri*, Δ4296, and Δ4296c strains on NB supplemented with sucrose 0.1% (*w*/*v*). (**B**) NB supplemented with sucrose 0.1% (*w*/*v*) and ampicillin 20 µg/mL. The figure shows the phase contrast, DAPI, and overlay of the two filters for *X. citri*; Δ4296 and Δ4296c. White arrows indicate the chromosome distribution, and red arrows indicate septum constriction (magnification of 100×)—scale = 5 µm.

**Figure 6 microorganisms-10-01008-f006:**
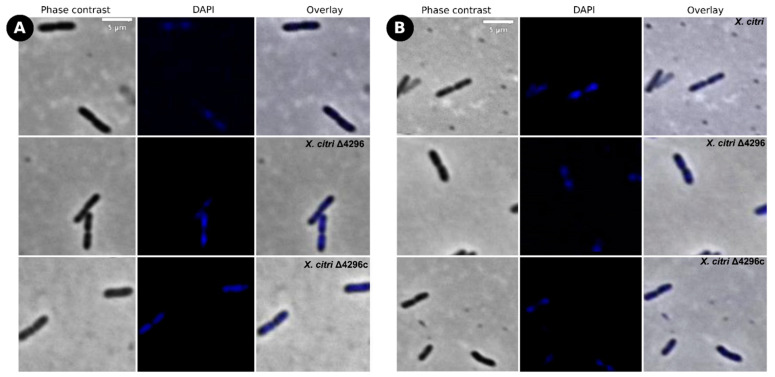
Short filaments, chain phenotype, and nucleoid organization show full reversion in glutamate’s presence as carbon source (**A**) Morphological analysis of *X. citri*, Δ4296, and Δ4296c strains on NB supplemented with glutamic acid 0.1% (*w*/*v*). (**B**) NB supplemented with glutamic acid 0.1% (*w*/*v*) and ampicillin 20 µg/mL. The figure shows the phase contrast, DAPI, and overlay of the two filters for *X. citri*, Δ4296, and Δ4296c (magnification of 100×)—Scale = 5 µm.

**Table 1 microorganisms-10-01008-t001:** Citrus canker quantification of “Pera Rio” orange leaves (*Citrus sinensis* L. Osbeck) by spray method. Strains were inoculated in a total of 65 leaves. The number of lesions was quantified on the abaxial surface of the leaves.

Strain	Number of Leaves with CC Lesions	Number of Leaves without CC Lesions	Total CC Lesions	Average of CC Lesions Per Leaf
*X. citri*	19	46	884 ^a^	46.52 ^a^
Δ4296	24	41	395 ^b^	16.45 ^b^

According to the Tukey test, different letters mean significant differences—*p* = 0.05.

**Table 2 microorganisms-10-01008-t002:** Statistics of morphotypes and nucleoid distribution of cells cultivated in rich medium (NB).

	Short Filaments %	Chain %	Nucleoid %
*X. citri*	0.50 ^a^	0 ^a^	0 ^a^
Δ4296	30.625 ^b^	15.75 ^b^	37.125 ^b^
Δ4296c	0.25 ^a^	0 ^a^	0.5 ^a^

Total *n* = 800 cells measured. Data correspond to the average cell morphology—different letters mean significant difference according to Welch’s ANOVA test- *p* = 0.05.

**Table 3 microorganisms-10-01008-t003:** Statistics of morphotypes and nucleoid distribution of cells cultivated in rich medium (NB) and ampicillin.

	Short Filaments (%)	Chain (%)	Nucleoid (%)
*X. citri*	0 ^a^	0.25 ^a^	0 ^a^
Δ4296	100 ^b^	50 ^b^	100 ^b^
Δ4296c	0 ^a^	0 ^a^	0 ^a^

Total *n* = 800 cells measured. Data correspond to the average cell morphology. According to Welch’s ANOVA test, different letters mean a significant difference—0.05.

**Table 4 microorganisms-10-01008-t004:** Short filaments, chain, and aberrant nucleoid organization phenotype were restored to normal on NB medium with sucrose or ampicillin and NB medium with glutamic acid or ampicillin.

		Short Filaments (%)	Chain (%)	Nucleoid (%)
NB medium sucrose 0.1%	*X. citri*	1 ^a^	0.38 ^a^	0 ^a^
	Δ4296	1.75 ^a^	0.75 ^a^	0.5 ^a^
	Δ4296c	0.25 ^a^	0.38 ^a^	0 ^a^
NB medium sucrose 0.1% ampicillin 20 µg/mL	*X. citri*	0.13 ^a^	1.25 ^a^	0,5 ^a^
	Δ4296	2.25 ^a^	1.88 ^a^	1 ^a^
	Δ4296c	0.25 ^a^	0.13 ^a^	0 ^a^
NB medium glutamic acid 0.1%	*X. citri*	0 ^a^	0 ^a^	0 ^a^
	Δ4296	0 ^a^	0 ^a^	0 ^a^
	Δ4296c	0 ^a^	0 ^a^	0 ^a^
NB medium glutamic acid 0.1% ampicillin 20 µg/mL	*X. citri*	0.13 ^a^	0 ^a^	0 ^a^
	Δ4296	0 ^a^	1 ^a^	0 ^a^
	Δ4296c	0 ^a^	0 ^a^	0 ^a^

Total *n* = 800 cells measured. Data correspond to the average cell morphology—same letters meaning no significant difference according to Welch’s ANOVA test- *p* = 0.05.

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
