# Peer review of "XAC4296 Is a Multifunctional and Exclusive Xanthomonadaceae Gene Containing a Fusion of Lytic Transglycosylase and Epimerase Domains"

_microorganisms, 2022, doi:10.3390/microorganisms10051008_

Round 1

Reviewer 1 Report

Overall the manuscript reads well and study is interesting. I Feel the manuscript is well presented and has the general flow. 

 It would be great if the microscopic data are supported with quantitative experiments. The microscopic images are not very clear. Do the authors have better images?

Author Response

Comments and Suggestions for Authors

Overall the manuscript reads well and the study is interesting. I Feel the manuscript is well presented and has a general flow.

It would be great if the microscopic data are supported with quantitative experiments. The microscopic images are not very clear. Do the authors have better images?

Quantitative experiments support all the microscopic data. This is shown in Tables 3 and 4. A total of N=800 cells were measured and used in the statistics

We also improved all the figures. We also submitted high-quality figures to the MDPI editorial office for proper manuscript production.

X. citri is a small cell, and unfortunately, it appears pretty thin under the microscope. Having a size that is a little more than the size of a visible light wave, X. citri is tricky to be observed in wide-field microscopy. And yet, we are confident that our images are clear in the sense that they show irrefutably the phenotypes we describe. We have improved the figures' resolution to minimize the sometimes blurry effect that wide-field can generate in thin/small cells. We hope it is clearer now.

Reviewer 2 Report

This paper describes the functionality of XAC4296, a multienzyme protein from Xanthomonas citri with Transglycosylase and Epimerase domains. A Δ4296 mutant can be supplemented with sucrose or glutamic acid as a carbon and nitrogen source, so the epimerase domain has been related to the central carbon and cell wall metabolism. The mutant also shows alterations in chromosome segregation and cell division.

It is an interesting article and I recommend its acceptance with a minor revision which I detail below.

Line 24 and Fig S3, it is stated that then gene AC4296 is expressed during plant-pathogen interaction. have other conditions been tested? would it be a constitutive gene?

Line  312. In Table 1 the average of CC per leaf wouldn't it be 395/24 = 16.45?

Please change commas to periods in Table 1, lines 320, 327 etc

Line 349, This is Table 2 right?

Fig S3 There are many differences among lanes 2 and 3 and lane 4 ,why? The lanes 2, 3 and 4 correspond to RNA from plant-pathogen interaction, doesn't it? It is not clear to me why it is said to be the expression of the XAC4296 gene in vitro.

In Fig 3 ,4 and S6, the DAPI staining looks bad, could these images be improved?

Line 352 and line 467, If XAC4296 is required for cell wall synthesis, why are there no differences in the growth shown in Fig S5?

Line 420, XAC4296 could involved in cell surface, is there any difference in plate colony appearance or biofilm production of the mutant compared to the wt strain?

Line 432, the dot is left over, and I think the right thing to write is "genera "no genus

Line 438,  the same, locus no loci

In Fig S1 is written Xanthomonas axonopodis strain 306, wouldn't it be Xanthomonas citri ?

No legend has been found for FigS6 and S7

Author Response

Comments and Suggestions for Authors

This paper describes the functionality of XAC4296, a multienzyme protein from Xanthomonas citri with Transglycosylase and Epimerase domains. A Δ4296 mutant can be supplemented with sucrose or glutamic acid as a carbon and nitrogen source, so the epimerase domain has been related to the central carbon and cell wall metabolism. The mutant also shows alterations in chromosome segregation and cell division.

It is an interesting article and I recommend its acceptance with a minor revision which I detail below.

Line 24 and Fig S3, it is stated that then gene AC4296 is expressed during plant-pathogen interaction. have other conditions been tested? would it be a constitutive gene?

We have tested the XAC4296 under plant-pathogen interaction (inoculated leaves) and in vitro conditions using antibiotics. Although our results may indicate that XAC4296 is expressed in non-infecting and infecting conditions, the bacteria are still viable when the XAC4296 is deleted. Therefore, it strongly suggests that XAC4296 is not a constitutive gene.

Line 312. In Table 1 the average of CC per leaf wouldn't it be 395/24 = 16.45?

Thanks for pointing this. We have corrected table 1

Please change commas to periods in Table 1, lines 320, 327 etc

corrected

Line 349, This is Table 2 right?

Yes, we have corrected the table caption

Fig S3 There are many differences among lanes 2 and 3 and lane 4 ,why? The lanes 2, 3 and 4 correspond to RNA from plant-pathogen interaction, doesn't it? It is not clear to me why it is said to be the expression of the XAC4296 gene in vitro.

We have added the Figure S3 legend for clarification.

The lanes 2,3,4 are replicates. The differences in the gel bands are related to the distinct concentration used in each lane.

Figure S3 Legend:

XAC4296 expression. 1% agarose gels showing expression of the XAC4296 in X. citri (M) 1Kb Fermentas DNA Ladder marker. (1) PCR product from genomic DNA from X. citri (~2400 bp) used as a positive control (non-infecting condition). (2,3,4) cDNA from X. citri representing the expression of XAC4296 in vitro.

In Fig 3 ,4 and S6, the DAPI staining looks bad, could these images be improved?

We have improved all figures

X. citri is a small cell, and unfortunately, it appears pretty thin under the microscope. Having a size that is a little more than the size of a visible light wave, X. citri is tricky to be observed in wide-field microscopy. And yet, we are confident that our images are clear in the sense that they show irrefutably the phenotypes we describe. We have improved the figures' resolution to minimize the sometimes blurry effect that wide-field can generate in thin/small cells. We hope it is clearer now.

Line 352 and line 467, If XAC4296 is required for cell wall synthesis, why are there no differences in the growth shown in Fig S5?

The phenotype is certainly not lethal, as the mutant grows, and maybe other enzymes can contribute to wall synthesis in a redundant function. Moreover, the fact that it forms filamentous cells may indicates a significant change in cell division that it is probably related to wall synthesis due to the increased sensitivity to ampicillin. We have improved this paragraph in the revised manuscript.

Line 420, XAC4296 could involved in cell surface, is there any difference in plate colony appearance or biofilm production of the mutant compared to the wt strain?

We did these analyses and did not find any difference in plate colony appearance or biofilm production of the mutant compared to the wt strain. We have added the following paragraph at the discussion:

Interestingly, we did not find any difference in plate colony appearance or biofilm production of the Δ4296 mutant compared to the X. citri wild type strain, indicating that, in principle, the XAC4296 epimerase domain is not related to cell surface polysaccharides production (data not shown). Indeed, the presented results indicated that the XAC4296 epimerase domain may be associated with other metabolic pathways, such as those related to bacterial cell wall metabolism.”

Line 432, the dot is left over, and I think the right thing to write is "genera "no genus

corrected

Line 438, the same, locus no loci

corrected

In Fig S1 is written Xanthomonas axonopodis strain 306, wouldn't it be Xanthomonas citri ?

Exactly. However, the figure was generated using the Integrated Microbial Genomes & Microbiomes (IMG/M) system comparative genomics analyses tool, relying on previous microorganisms nomenclature.

No legend has been found for FigS6 and S7

We have added all supplementary material legends